Dimensional changes of commercial and novel polyvinyl siloxane impression materials following sodium hypochlorite disinfection

Ud Din Shahab 1 drshahab728@hotmail.com
Sajid Muhammad 2
Saeed Asfia 2
http://orcid.org/0000-0002-1748-0413 Chaudhary Farooq Ahmad 1
http://orcid.org/0000-0001-7131-1752 Alam Mohammad Khursheed 3
Sarfraz Juneda 1
Ahmed Bilal 1
Patel Mangala 4
1 School of Dentistry (SOD), Federal Medical Teaching Institution (FMTI)/PIMS, Shaheed Zulfiqar Ali Bhutto Medical University (SZABMU) , Islamabad , Pakistan
2 Department of Dental Materials, Islamabad Medical & Dental College , Islamabad , Pakistan
3 Preventive Dentistry Department, College of Dentistry, Jouf University , Sakaka , Saudi Arabia
4 Centre for Oral Bioengineering (Dental Physical Sciences Unit), Bart’s and The London School of Medicine and Dentistry, Queen Mary University of London , London , United Kingdom
Williams John Leicester
Electronic publication date: 2022 Jan 28
Publication date: 2022
Volume: 10
Electronic Location ID: e12812
Received 2021 Aug 23; Accepted 2021 Dec 29
Copyright: © 2022 Ud Din et al.
Copyright year: 2022
Copyright holder: Ud Din et al.
License: This is an open access article distributed under the terms of the Creative Commons Attribution License, which permits unrestricted use, distribution, reproduction and adaptation in any medium and for any purpose provided that it is properly attributed. For attribution, the original author(s), title, publication source (PeerJ) and either DOI or URL of the article must be cited.
License URL: https://creativecommons.org/licenses/by/4.0/

Keywords: Impression materials, Vinyl polysiloxane, Disinfection, Sodium hypochlorite, Dimensional changes

Funding: The authors received no funding for this work.

==============================
Background

Dental impressions are used to record anatomy of teeth and surrounding oral structures. Impression materials become contaminated with saliva and blood requiring disinfection, which may have negative impact on dimensional stability of materials.

Objective

Comparatively evaluate linear dimensional changes of synthesized Tetra-functional (dimethylsilyl) orthosilicate (TFDMOS) containing Polyvinylsiloxane (PVS) impressions following sodium hypochlorite disinfection.

Methods

Percentage dimensional changes of three commercial PVS (Elite HD Monophase, Extrude and Aquasil Ultra Monophase) and five experimental PVS impression materials were measured. Experimental material contained novel cross-linking agent (TFDMOS) and a non-ionic surfactant (Rhodasurf CET-2) that is Exp-A (without TFDMOS), Exp-B (with TFDMOS), Exp-C (TFDMOS+ 2% Rhodasurf CET-2), Exp-D (TFDMOS+ 2.5% Rhodasurf CET-2) Exp-E (TFDMOS+ 3% Rhodasurf CET-2). Samples were made using rectangular stainless-steel molds (40 × 10 × 3 mm3) and linear dimensional changes were measured using a calibrated travelling microscope at 10× magnification after immersion in distilled water (D.W) and 1% Sodium Hypochlorite solution at two different time intervals i.e., 30 min and 24 h.

Results

Samples immersed in 1% NaOCl showed significant (p < 0.05) dimensional changes after 30 min of immersion. Exp-E showed significantly greater dimensional changes than their control (Exp-A and Exp-B). In distilled water, there were no significant difference among the tested materials. Aquasil exhibited highest expansion (0.06%) in both solutions. At 24 h, among the commercial materials, Extrude had the greatest expansion followed by Aquasil and Elite in DW while Aquasil showed the greatest expansion followed by Extrude and Elite in NaOCl.

Conclusion

Experimental PVS had linear dimensional changes within the ISO 4823; 2015 recommended range. However, extended immersion can negatively affect the linear dimensions.

Introduction

Dental impressions are used to accurately record and reproduce the shape, relationship of teeth with surrounding oral structures (Kumari & Nandeeshwar, 2015). However, during impression making, the material becomes contaminated with saliva and often with blood which may pose health hazard to the dental team (Ud Din et al., 2018). Different methods of disinfection, including spraying and immersion of impression material are commonly employed (Khan & Mushtaq, 2018). However, the American Dental Association (ADA) recommends the immersion technique because it allows direct contact of disinfectant solutions with all surfaces of the impression (Samra & Bhide, 2018). Disinfection through immersion reduces the risk of cross infection, yet alteration in dimensions of impression materials and negative impact on quality of the resulting cast has often been reported (Guiraldo et al., 2012; Martin, Martin & Jedynakiewicz, 2007). This might result in dimensional changes in dental cast prosthesis and ultimately affect the fit of the final restoration (Asopa et al., 2020). Iwasaki et al. (2016) reported marked changes in the dimensions of alginate impressions due to water sorption after prolonged immersion in a disinfectant solution (Iwasaki et al., 2016). Similarly, Babiker, Khalifa & Alhajj (2018) reported significant dimensional changes of gypsum casts obtained from irreversible hydrocolloids after disinfection by NaOCl immersion (Babiker et al., 2018).

The recommended products for disinfection of the impression materials are sodium hypochlorite, chlorhexidine, glutaraldehyde and iodine agents (Correia-Sousa et al., 2013). Among numerous disinfectants available, sodium hypochlorite has been recommended by the ADA and Environmental Protection Agency (EPA) for the disinfection of impressions due to its low cost and broad spectrum antimicrobial activity (Correia-Sousa et al., 2013; Khan & Mushtaq, 2018). A study by Al-Enazi and Naik on the efficacy of 1% sodium hypochlorite and 2% glutaraldehyde disinfectant sprays on impression materials (alginate and addition silicone rubber-based impression material) established that 1% sodium hypochlorite yielded better results (Al Enazi & Naik, 2016).

Polyvinylsiloxane is popular due to its ability to precisely record tissue details, excellent physical properties and easy manipulation, and is preferred for making final impressions of fixed dental prosthesis (Azevedo et al., 2019). Nevertheless, these materials have lower tear strength and percentage elongation compared to polysulphide. Additionally, the hydrophobic nature of the material demands a dry field of operation to accurately record fine tissue details (Ud Din et al., 2018).

Different researchers have incorporated surfactant to the material composition to synthesize hydrophilic addition silicones, but limited literature is available on methods to improve the tear strength of the material (Lee et al., 2004). Formerly, we have carried out synthesis of novel TFDMOS containing polyvinyl siloxane with a non-ionic surfactant (Rhodasurf CET-2) to improve the tear strength and hydrophilicity of conventional addition silicones. It was observed that experimental formulation had significantly higher tear strength compared to commercial material and that tear strength was directly proportional to the concentration of non-ionic surfactant (Din et al., 2018). Considering the variation in material properties corresponding to the composition of the material, the effect of non-ionic surfactant on dimensional stability of materials especially after disinfection needed further exploration. It was hypothesized that the experimental formulations have higher water sorption due to the presence of non-ionic surfactant, resulting in noticeable dimensional changes.

Extensive use of polyvinylsiloxane in dentistry and varieties of formulation available necessitated a comprehensive comparison of material particularly in terms of dimensional changes following long term disinfection. Therefore, the objective of this study was two-fold including formulation of experimental polyvinylsiloxane containing TFDMOS and evaluation of the influence of storage media that is distilled water and 1% sodium hypochlorite on the linear dimensional changes over the period of 24 h, in commercial and synthesized PVS impression materials.

Materials and Methods

In this in-vitro experimental study, three commercially available medium bodied PVS materials, Aquasil (Ultra, Dentsply, PA, USA), Elite HD Monophase (Zhermack, Badia Polesine, Italy) and Extrude (Kerr, CA, USA) were selected. In addition, five experimental PVS materials were prepared as base and catalyst paste by weighing all components using a four beam balance (Mettler, Toledo Ltd, Model AG204, UK) (Ud Din et al., 2018). The composition of experimental materials is illustrated in Table 1. To prevent premature polymerisation, equal amount of the prepared material was weighed with a four beam balance and was packed into separate compartments of an auto-mixing cartridge and stored at 4 °C. Exp-A served as a control for Exp-B, while Exp-B was used as a control for Exp-C, D and E. Rhodasurf CET-2 (non-ionic surfactant) was incorporated in the base paste of Exp-B at 2, 2.5 and 3% to formulate Exp C, D and E respectively.

Table 1 Composition of Experimental Polyvinyl siloxane impression materials.

Components	Base paste (Wt %)	
Exp-A	Exp-B	Exp-C	Exp-D	Exp-E	
Vinyl-terminated dimethylpolysiloxane (Mw 62700)	39.90	39.90	37.95	37.46	36.98	
Polymethylhydrosiloxane (Mw 2270)	1.10	0.77	0.74	0.73	0.72	
Tetra-functional (dimethylsilyl) orthosilicate (Mw 329)	–	0.33	0.32	0.31	0.31	
Components	Catalyst paste (Wt %)	
Exp-A	Exp-B	Exp-C	Exp-D	Exp-E	
Vinyl-terminated dimethylpolysiloxane (Mw 62700)	40.72	40.72	39.51	39.51	39.51	
Platinum (0.05 M)	0.06	0.06	1.27	1.27	1.27	
Palladium (<1 µm)	0.23	0.23	0.22	0.22	0.22	
Rhodasurf CET-2	–	–	2.00	2.50	3.00	

A total of 80 samples (n = 10 for each material) were prepared by introducing the material from automixing syringe into the preformed stainless steel mold (40 × 10 × 2 mm3) sandwiched between two metal plates lined by acetate paper. The assembly was placed under a hydraulic press (MESTRA MOD-030350; Talleres Mestraitua, Bizkaia, Spain). Commercial material was allowed to polymerize for the time specified by the manufacturers, while experimental formulation were polymerized for 10 min (Ud Din et al., 2018; Din et al., 2018).

After polymerization, the material was removed from the mold and the specimens were divided into two test groups, each consisting of five samples from each material. Group 1 was immersed in 1% NaOCl and Group 2 in distilled water solution for 24 h at 23 ± 1 °C in an oven (Qualicool incubators; LTE Scientific Ltd, Oldham, UK). At predetermined time periods, which were 10 and 30 min (replicate at-office disinfection), and 60 min and 24 h (simulating the transportation time for the impression to reach the dental laboratory) each sample was removed from the respective liquid and blot dried. Linear dimensional changes were recorded using a calibrated travelling microscope (Chesterman, Sheffield, England) by measuring the distance between the fixed edge and free end of the sample (movable pin) at 10× magnification. Data were presented as mean and standard deviation using SPSS Version 22 (Armonk NY IBM Corp, Armonk, NY, USA). Analysis of variance with post hoc Tukey’s test was performed to statistically compare all PVS materials at different time points and p value of 0.05 was considered as significant.

Results

Figures 1 and 2 show percentage linear dimensional changes of impression materials in DW and 1% NaOCl after 24 h immersion. Aquasil exhibited the highest expansion (0.06%) in both solutions while Exp-A exhibited the least expansion (0.01%) in DW. For Group 1, a significant difference (p ≤ 0.05) in dimensions were observed after 30 min of immersion. Inter-group analysis revealed statistically similar dimensional changes among all tested materials, except for Exp-E (Table 2). The surfactant modified Exp-E showed significantly (p ≤ 0.05) greater dimensional changes than their control groups (Exp A & Exp B). Among commercial materials, although no significant changes in dimensions were observed, Aquasil showed the highest expansion followed by Extrude and Elite in 1% NaOCl solution.

Figure 1 Mean linear dimensional changes with Standard Deviation of commercilal and experimental PVS impression materials over the period of 24 h following immersion in Distal water.

Figure 2 Mean linear dimensional changes with Standard Deviation of commercilal and experimental PVS impression materials over the period of 24 h following immersion in 1% NaOCl.

Table 2 Comparison of linear dimensional changes of impression materials stored in 1% NaOCl disinfectant solution at 30 min.

Materails	Mean	S.D	F (p-value)	Post –hoc Tukey test	p-value	
Aquasil	0.058	0.029	3.657
(0.005)	Exp E vs Aquasil	0.634	
Extrude	0.046	0.027	Exp E vs Extrude	0.252	
Elite	0.021	0.028	Exp E vs Elite	0.012	
Exp A	0.010	0.023	Exp E vs Exp A	0.003	
Exp B	0.031	0.028	Exp E vs Exp B	0.047	
Exp C	0.041	0.023	Exp E vs Exp C	0.159	
Exp D	0.032	0.029	Exp E vs Exp D	0.053	
Exp E	0.040	0.023	

There were no significant differences among the commercial and experimental materials for Group 2 immersed in distilled water. Extrude had the highest expansion followed by Aquasil and Elite at 24 h. Exp B showed slightly higher, but statistically similar expansion to Exp-A, while Exp-C, D and E exhibited higher expansion in each immersion solution compared to Exp-B (control) at each interval. Also, expansion of experimental materials was directly related to the concentration of the surfactant.

Discussion

Impression making is an important aspect of prostheses fabrication. In the oral cavity, these materials come in contact with saliva and blood (Khan & Mushtaq, 2018). This demands disinfection of materials to prevent cross-infection in dental clinics and hospital. However, it is important that the impressions remain dimensionally stable after disinfection.

The results of the present study revealed that immersion of addition silicones impression materials in distilled water (Group 2) had no significant impact on the dimensional stability compared to disinfection of impression with 1% NaOCl (Group 1) for 24 h which resulted in expansion of the tested materials. However, the dimensional changes observed were not clinically relevant as the values were within the permitted range of ≤0.5% as recommended by ISO 4823 (2000); 2015 (ISO) and ADA specification 19 (Association ANSiAD, 2007); thus, indicating high precision of the experimental material.

The limited amount of linear expansion could be attributed to the isotropic expansion of the samples as they adhere to the PTFE trough of the travelling microscope. These findings were supported by Carvalhal et al. (2011) who investigated the dimensional changes of polysulphide, polyether, addition and condensation silicone after immersion disinfection with 0.5% NaOCl and 2% glutaraldehyde. They observed insignificant changes in the linear dimensions of the materials and recommended that all synthetic elastomers can be safely disinfected by immersion in 0.5% NaOCl and 2% glutaraldehyde (Carvalhal et al., 2011). In a similar study, Samra & Bhide (2018) observed clinically acceptable dimensional changes for alginate and PVS after disinfection with ultraviolet rays and immersion disinfection with sodium hypochlorite, but changes in cross-arch space and inter-abutment distance on gypsum casts were noted for materials disinfected by glutaraldehyde immersion (Samra & Bhide, 2018).

Furthermore, Exp-E had significantly greater dimensional changes after immersion disinfection with NaOCl. This was in line with a number of studies that indicated hydrophilic silicones have greater tendency to absorb water and expand (Silva & Salvador, 2004). Results also indicated that longer immersion periods can affect the materials. These findings were similar to those noted by Johnson (1986), who reported long-term dimensional stability of PVS to be unsatisfactory (Thouati et al., 1996). Similarly, Kumari & Nandeeshwar (2015) observed significant difference between non-disinfected and disinfected specimens of PVS and polyether after immersion of 16 h in 0.525% sodium hypochlorite (Kumari & Nandeeshwar, 2015). On contrary, Nassar et al. (2017) found no statistically significant difference between dimensions of vinyl polyether silicone at the time of fabrication and after storage for 1 and 2 weeks in 2.5% buffered solution of glutaraldehyde and reported that dimensional changes within the material obeyed ANSI/ADA standards.

The limitation of this study was that the accuracy of the impression material was determined by studying the stability of the impression itself and that the setting expansion of the gypsum cast was not taken into account. However, the results confirmed the high precision of the experimental materials, suggesting routine use of these materials for impression of fixed and removable partial dentures, complete dentures, precision attachments and implants. However, it is suggested to evaluate the effects of other disinfectant systems.

Conclusions

All three commercial and five experimental polyvinylsiloxane impressions materials exhibited linear expansion within the recommended range outlined by ANSI and ISO 4823 (2000); 2015 following 24 h of immersion, though extended immersion time amplified the dimensional changes within all the groups. Therefore, following disinfection, the silicone impression materials must be poured within 24 h.

Supplemental Information

Supplemental Information 1 Raw data.

The raw data collected during the experimental work in the laboratory for all commercial and experimental materials used in this study including individual files for all materials separately and a combined file after analysis the data along with the graphic representation.arrahcs/EvoloPy-FS.

Click here for additional data file.

Additional Information and Declarations

Competing Interests

Author Contributions

Data Availability

Mohammad Khursheed Alam is an Academic Editor for PeerJ.

Shahab Ud Din conceived and designed the experiments, performed the experiments, analyzed the data, prepared figures and/or tables, authored or reviewed drafts of the paper, and approved the final draft.

Muhammad Sajid performed the experiments, prepared figures and/or tables, and approved the final draft.

Asfia Saeed performed the experiments, prepared figures and/or tables, and approved the final draft.

Farooq Ahmad Chaudhary conceived and designed the experiments, analyzed the data, prepared figures and/or tables, authored or reviewed drafts of the paper, and approved the final draft.

Mohammad Khursheed Alam analyzed the data, prepared figures and/or tables, and approved the final draft.

Juneda Sarfraz analyzed the data, prepared figures and/or tables, and approved the final draft.

Bilal Ahmed analyzed the data, authored or reviewed drafts of the paper, and approved the final draft.

Mangala Patel conceived and designed the experiments, performed the experiments, authored or reviewed drafts of the paper, and approved the final draft.

The following information was supplied regarding data availability:

The raw data is available in the Supplemental File.

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
