# Peer review of "Dimensional changes of commercial and novel polyvinyl siloxane impression materials following sodium hypochlorite disinfection"

_PeerJ, doi:10.7717/peerj.12812_

## Round 0.1 · original submission · Minor Revisions

Please remove the first sentence from the Discussion section. The reviewers have requested further information be included about the statistics in the methods and results sections.

Reviewer 1 ·

Basic reporting

Please expand more on the introduction of why it is important to conduct this study and in particular, why is dimensional changes important and what implication can it be lead to if it is not achieved.

Statement of the problem and justification of study need to be enhanced in introduction section. Why use 1% sodium hypoclorite in this study (and what other solutions is used commonly)

Experimental design

ok

Validity of the findings

ok

Additional comments

number of whole sample size should be included and written clearly
Line 108 n=10 is refering to which group?
Line 123, to add statistical significance

Why do you choose 30 mins to check for dimensional stability - add reference or justify
Line 145- Remove add discussion here
Line 147- correct quoting of referencing

·

Basic reporting

1.Few grammar mistakes are there. I have highlighted them and added questions as per the sticky notes in the content of manuscript where there is doubt.
2. Authors are requested to go through all the highlighted content and make corrections. ADA specifications number needs to be added.
3. Clarify that you are testing for final impressions for fixed prosthesis in your material & Methods.
4. Post hoc tukey test table is missing.

Experimental design

1. Explain four figure balance briefly.
2. How was polymerisation of specimens checked.
3. Mention being an in vitro study in Material & Methods.
4. Tables to be included of multiple comparisons. That's vital and needs to be cited in text later.

Validity of the findings

Nicely written except for few changes in grammar that have been highlighted.
2. Legends under the figure needs to be corrected.

Reviewer 3 ·

Basic reporting

The submitted manuscript reports experimental data on impression materials treated with sodium hypochlorite disinfection, a very interesting topic, especially during these times.
However, some flaws do not allow me to give a green light for acceptance, but changes are required to improve the quality of the manuscript, to make it more appealing, readable and therefore to contribute fully to clinicians’ practice.

Experimental design

Let’s start from the abstract:
1) It would be better to specify in the methods what You did and what You used: Rodhasurf indeed it is not specified for example.
2) In results, Extrude should be with capitol letter and DW should be specified before when You name distilled water
Introduction
Line 57: of impression is repeated
Lines 84-88: literature reference is missing
Methods
Line 108: eighty sample (n 10) numbers does not match, or it seems confusing to the reader. Please rephrase it clearly
Lines 115-120: the paragraph must be rewritten due to the presence of several grammar and punctuation errors

Validity of the findings

Statistics: have You checked the distribution before to perform the ANOVA?
Results: it would be nice to add the microphotographs of the sample
Discussion
Line 145: add Your discussion here must be deleted.
Lines 154-155: the sentence must be rewritten due to the presence of several grammar and punctuation errors
Line 162-166: the paragraph must be rewritten due to the presence of several grammar and punctuation errors
Line 174: the full stop should be placed after the round bracket and after “et al”

The methods are appropriate and well described, and adequate details are provided to replicate the work.

In the discussion, the most important part of the article, its core, some very important concepts are mentioned but not fully and properly discussed.
Indeed, an accurate impression means an accurate prosthetic rehabilitation (I might suggest this article to stress out this concept Traini, Tonino et al. “Esthetic outcome of an immediately placed maxillary anterior single-tooth implant restored with a custom-made zirconia-ceramic abutment and crown: a staged treatment.” Quintessence international (Berlin, Germany: 1985) vol. 42,2 (2011): 103-8.)
Another concept that instead totally lacks is the role of the innovation technology, from the reproducibility performance to the digital format in achieving the ideal “material and methods” to realize the most precise prosthetic rehabilitation (I might suggest these articles to stress out this concept Varvara, G et al. “Comparative surface detail reproduction for elastomeric impression materials: Study on reproducibility performance.” Journal of biological regulators and homeostatic agents vol. 35,1 (2021): 161-169. doi:10.23812/20-561-A, Turkyilmaz, Ilser et al. “Guest Commentary: The Battle of File Formats from Intraoral Optical Scanners.” The International journal of prosthodontics vol. 33,4 (2020): 369-371. doi:10.11607/ijp.2020.4.gc).

---

## Round 0.2 · Minor Revisions

Your article will be accepted following minor corrections. Please make the editorial corrections as suggested in the attached edited copy of the manuscript.

·

Basic reporting

1. Most of the changes have been implemented.
2. Slight correction required in a line in discussion which has been edited by me in the file.
3. Commercial spelling wrong in the legend for graphs at 3 places.

Experimental design

All corrections have been implemented.

Validity of the findings

1. In the end, significant 'p value' of post hoc Tukey test should be written.

Additional comments

Article can be accepted after these minor corrections.

---

## Round 0.3 · accepted · Accept

Thank you for revising you manuscript and making all the suggested changes.